# Compound C Inhibits Renca Renal Epithelial Carcinoma Growth in Syngeneic Mouse Models by Blocking Cell Cycle Progression, Adhesion and Invasion

**DOI:** 10.3390/ijms23179675

**Published:** 2022-08-26

**Authors:** Myungyeon Lee, Na Yeon Ham, Chi Yeon Hwang, Jiwon Jang, Boram Lee, Joo-Won Jeong, Insug Kang, Eui-Ju Yeo

**Affiliations:** 1Department of Medicine, Gachon University School of Medicine, Incheon 21999, Korea; 2Department of Biochemistry, College of Medicine, Gachon University, Incheon 21999, Korea; 3Department of Biomedical Sciences, Graduate School, Kyung Hee University, Seoul 02447, Korea; 4Department of Anatomy and Neurobiology, School of Medicine, Kyung Hee University, Seoul 02447, Korea; 5Department of Biochemistry and Molecular Biology, Biomedical Science Institute, School of Medicine, Kyung Hee University, Seoul 02447, Korea

**Keywords:** Compound C, renal cancer cells, Renca cells, BALB/c, ROS, G2/M cell cycle arrest

## Abstract

Compound C (CompC), an inhibitor of AMP-activated protein kinase, reduces the viability of various renal carcinoma cells. The molecular mechanism underlying anti-proliferative effect was investigated by flow cytometry and western blot analysis in Renca cells. Its effect on the growth of Renca xenografts was also examined in a syngeneic BALB/c mouse model. Subsequent results demonstrated that CompC reduced platelet-derived growth factor receptor signaling pathways and increased ERK1/2 activation as well as reactive oxygen species (ROS) production. CompC also increased the level of active Wee1 tyrosine kinase (P-Ser^642^-Wee1) and the inactive form of Cdk1 (P-Tyr^15^-Cdk1) while reducing the level of active histone H3 (P-Ser^10^-H3). ROS-dependent ERK1/2 activation and sequential alterations in Wee1, Cdk1, and histone H3 might be responsible for the CompC-induced G2/M cell cycle arrest and cell viability reduction. In addition, CompC reduced the adhesion, migration, and invasion of Renca cells in the in vitro cell systems, and growth of Renca xenografts in the BALB/c mouse model. Taken together, the inhibition of in vivo tumor growth by CompC may be attributed to the blockage of cell cycle progression, adhesion, migration, and invasion of tumor cells. These findings suggest the therapeutic potential of CompC against tumor development and progression.

## 1. Introduction

Renal cell carcinoma (RCC) is the most common and deadliest type of kidney cancer with a mortality rate of 30–40% [1]. Its incidence has increased rapidly in recent decades [2]. The risk factors include genetic as well as epigenetic alterations [3,4], sex (males more than females), environmental conditions, and lifestyle factors [5] such as obesity, hypertension, smoking, or long-term dialysis due to chronic kidney disease [1,6,7]. Although surgical resection, such as nephrectomy, is the main treatment strategy, adjuvant and targeted therapies are also used in patients with metastatic RCC. Since most metastatic RCCs are resistant to chemotherapy and radiotherapy, new biomarkers and drug candidates are needed for early detection and treatment [8,9,10].

AMP-activated protein kinase (AMPK) is an evolutionarily conserved energy sensor important for regulating cell growth, proliferation, survival, and metabolism [11,12]. Active AMPK inhibits biosynthetic enzymes, allowing cells to retain essential nutrients and energy during metabolic crisis [12,13]. Despite our knowledge of this important kinase, there are no specific chemical inhibitors available to investigate its function. However, Compound C (CompC), a small molecule, has been widely used in cell-based, biochemical, and in vivo assays as an AMPK inhibitor.

In a previous study, we reported that CompC reduced cell viability in human diploid fibroblasts (HDFs) and other cells that express platelet-derived growth factor receptors (PDGFR) by inhibiting PDGF-induced signaling pathways, including PDGFR tyrosine kinase, PI3K, PLCγ1, ERK1/2, and Akt in an AMPK-independent manner [14]. Its cellular effects have also been attributed to its inhibitory action on cell cycle regulatory proteins such as phosphorylated pRB, cyclins, cyclin-dependent protein kinases (Cdks), and Cdk inhibitors [14,15]. CompC inhibits cell proliferation via cell cycle arrest at the G0/G1 phase in HDFs [14], and at the G2/M phase in B16F1 melanoma cells [15]. According to previous reports, the AMPK-independent pathway includes inhibition of mTORC1/C2, activation of calpain/cathepsin pathway, as well as induction of necroptosis and autophagy [12].

Several studies have confirmed the potential of activated AMPK to promote apoptosis [16,17]. AMPK activation is often accompanied by increased levels of reactive oxygen species (ROS). Interestingly, the pro-apoptotic action of the AMPK inhibitor CompC has also been reported to involve induction of oxidative stress [18]. In addition, it reportedly inhibits hypoxia-inducible factor-1α (HIF-1α) activation via a mechanism independent of AMPK, but dependent on cellular oxygen availability through interaction with the mitochondrial electron transport chain [19,20].

Metastasis is the main cause of high mortality and poor clinical outcomes associated with cancer [21]. It consists of several interdependent processes, including the uncontrolled proliferation, adhesion, migration, and invasion of cancer cells of distant sites in surrounding tissues and organs [22,23]. A previous study reported that CompC inhibits the proliferation and migration of human umbilical vascular endothelial cells (HUVECs) via inhibition of vascular endothelial growth factor receptor (VEGFR)-induced signaling pathways [15]. It also reduces the anti-invasive effects of fenofibrate, which is an agonist of peroxisome proliferator-activated receptor alpha (PPARα) and acts as a lipid-lowering agent, in CAL 27 human oral cancer cells [24]. This raises the question of whether CompC has any effect on the adhesion and invasion of Renca cells.

Since our preliminary experiments revealed that CompC reduces the viability of Renca murine renal epithelial carcinoma cells, the molecular mechanism underlying its anti-proliferative effect was investigated and its effect on cell cycle progression was analyzed by flow cytometry. In addition, to confirm how CompC induces cell cycle arrest in Renca cells, its effect on the expression and phosphorylation status of signaling as well as cell cycle regulatory proteins was examined by western blot analysis. We also investigated its effect of CompC on the adhesion, migration, and invasion of Renca cells. Renca xenografts were generated in syngeneic BALB/c murine models and the effect of CompC on in vivo tumor growth was evaluated.

## 2. Results

### 2.1. CompC Reduces the Viability of Renca Cells via G2/M Cell Cycle Arrest

The effect of CompC on cell viability was determined in various RCCs including Renca, A498, Caki-1, and Caki-2 cells. Cells were treated with the vehicle dimethyl sulfoxide (DMSO) or various concentrations (1, 2.5, 5, 10, and 20 μM) of CompC for two days in the culture medium [DMEM with 10% fetal bovine serum (FBS)]. Cell viability was measured using the 3-(4,5-dimethylthiazol-2-yl)-2,5-diphenyltetrazolium bromide (MTT) assay. The results showed that the viability of various mouse and human RCCs treated with 1–20 μM CompC decreased in a dose-dependent manner (Figure 1a). As shown in Figure 1a, the cytotoxic effect of CompC was slightly greater in Renca cells than in other RCCs. The half maximal inhibitory concentrations (IC_50_) of CompC for the inhibition of each RCC after two days were determined to be 4.3 (Renca), 6.4 (A498), 6.8 (Caki-1), and 6.4 μM (Caki-2). These results suggest that CompC is more effective against Renca cells than against other RCCs. Among the various animal models for spontaneous renal carcinoma [25], BALB/c, a syngeneic mouse model for Renca, was available in our country. Renca cells were selected for further studies on the effect of CompC on RCCs.

To determine whether the observed inhibitory effect of CompC on the viability of RCCs was caused by cell cycle arrest or apoptotic cell death, morphological changes in the cells were examined. Renca cells and other RCCs were treated with the vehicle or 10 μM CompC for one–two days in the culture medium while their morphological changes were observed and photographed using an inverted microscope. CompC-treated Renca cells did not show any visible apoptotic cell death after one–two days, as compared to the vehicle-treated cells in the presence of 10% FBS (Figure 1b).

To confirm whether CompC inhibits cell proliferation by cell cycle arrest, DNA content of the cells was evaluated by flow cytometry after staining the nuclei with propidium iodide (PI). For the experiment, Renca cells were serum starved by incubation with a serum-free medium (SFM) containing 0.1% bovine serum albumin (BSA) overnight. After pretreatment with the vehicle (DMSO) or 10 μM CompC for 1 h, the cells were stimulated with serum (10% FBS) for the indicated times (8–24 h). The data showed that CompC increased the number of cells in the G2/M fraction from 32.4 to 34.9% at 8 h, from 26.7 to 57.8% at 16 h, and from 34.6 to 65.3% at 24 h (Figure 2). On the other hand, it reduced cells in the G1 phase from 37.8 to 38.0% at 8 h, from 39.3 to 9.2% at 16 h, and from 37.7 to 15.2% at 24 h and those in the S phase from 22.5 to 20.2% at 8 h, from 25.4 to 18.2% at 16 h, and from 20.1 to 10.2% at 24 h (Figure 2). These results suggest that CompC inhibits cell proliferation by inducing G2/M cell cycle arrest and reducing cell cycle progression to the G1 and S phases through mitosis in Renca cells.

There was only a slight increase in apoptotic cell death, as proven by the altered sub-G1 fraction in CompC-treated cells during the experimental periods: 7.3% to 6.9% at 8 h, 8.6% to 14.9% at 16 h, and 7.6% to 9.3% at 24 h (Figure 2). Apoptosis was also proven by Annexin V/PI double staining and only a slight increase (2–6%) was observed in the late apoptotic fraction of CompC-treated cells compared to vehicle-treated cells (data not shown). Since the apoptotic insults on Renca cells induced by CompC were relatively weak in our experimental conditions, the present study focused on the molecular mechanism of G2/M cell cycle arrest.

### 2.2. CompC Induces G2/M Cell Cycle Arrest via ROS Production in Renca Cells

To elucidate the mechanism by which CompC affects cell cycle progression in Renca cells, its effect on the expression and phosphorylation status of signaling and cell cycle regulatory proteins was examined by western blot analysis. As the inhibitory short-term effect of CompC on PDGF-induced cell signaling has been suggested in a previous study [14], the effect of CompC on PDGFR tyrosine kinases and the downstream signaling protein Akt was examined after PDGF treatment. Subconfluent Renca cells were serum starved overnight and pretreated with the vehicle or 10 μM CompC for 1 h. PDGF was then added to the cells and incubated for the indicated times (1–30 min). As shown in Figure 3a, CompC blocked the PDGF-induced phosphorylation of PDGFRβ and Akt in Renca cells. The effect of CompC on FBS-induced PDGFR tyrosine kinases and downstream signaling proteins, including Akt, ERK1/2, PI3K, and PLCγ1, was also examined. The results showed that CompC treatment reduced the activation/phosphorylation of PDGFR and the downstream signaling proteins, Akt, PI3K and PLCγ1, in Renca cells (Figure 3b). In contrast, the level of phosphorylated ERK1/2 was enhanced by the serum treatment, and was higher in CompC-treated cells than in vehicle-treated cells at 5–10 min (Figure 3b). These results suggest that the effect of CompC on cell cycle arrest may be mediated by an imbalance among signaling proteins, such as Akt and ERK1/2, in Renca cells.

ROS have been shown to play a role in ERK activation [15,26]. ROS-induced ERK activation has also been implicated in cell cycle arrest [27,28]. Therefore, to confirm that CompC-induced ERK1/2 activation might be due to ROS production in Renca cells, we measured the ROS levels in vehicle- or CompC-treated cells. The level of intracellular ROS was evaluated by flow cytometry after treatment of the cells with dichlorodihydrofluorescein diacetate (DCF-DA). The results demonstrated that CompC treatment significantly elevated ROS levels at 1 and 6 h in cells cultured in SFM and FBS medium (Figure 4a,b). Higher ROS production was observed in the FBS medium than in the SFM medium. Pretreatment of cells with 5 mM N-acetylcysteine (NAC) reduced CompC-induced ERK1/2 activation, as determined by western blot analysis (Figure 4c). These data suggested that CompC-induced ERK1/2 activation may be caused in part by ROS production. Furthermore, to explain the role of CompC-induced ROS production and ERK1/2 activation in G2/M cell cycle arrest, the serum-stimulated cell cycle was examined by flow cytometry after the pretreatment of cells with the vehicle DMSO or NAC for 1 h, followed by treatment with the vehicle or CompC. Treatment with CompC alone resulted in a significant increase (*** *p* < 0.001) in G2/M-arrested cells, and the gated percentage of G2/M-arrested cells was significantly (^#^ *p* < 0.05) reduced by pretreatment with NAC at 16 and 24 h (Figure 4d). In contrast, NAC slightly, but without a significant difference, increased the percentage of gated fractions of G1 and S at 16 h compared to the CompC alone-treated cells (Figure 4e). The MTT assay also showed that treatment with NAC significantly (^##^ *p* < 0.01) increased cell viability compared to cells treated with CompC alone (Figure 4f). Altogether, these results suggest that ROS production may play a role in CompC-induced G2/M cell cycle arrest in Renca cells.

### 2.3. CompC Inhibits the Phosphorylation of Histone H3 via the Inhibitory Tyrosine Phosphorylation of Cdk1 by Wee1

To clarify how CompC induces G2/M cell cycle arrest in Renca cells, its long-term effect on the expression and phosphorylation status of cell cycle-related proteins was examined using western blot analysis. Subconfluent Renca cells were serum starved overnight, pre-treated with the vehicle or 10 μM CompC for 1 h, and then stimulated with 10% FBS for 8–24 h.

Cdk1, also known as Cdc2, functions as a serine/threonine kinase that plays an important role in the entry of cells into mitosis [29]. Cdk1 complexes with cyclin B1 (or cyclin A) and phosphorylates a variety of target substrates, including histone H3, another mitotic indicator. Cdk1 phosphorylation at Thr^161^ by the Cdk-activating kinase (CAK) induces activation of Cdk1, whereas phosphorylation at Tyr^15^ induces its inhibition [29,30]. Interestingly, we observed that CompC treatment increased the level of phosphorylated Cdk1 on Tyr^15^ (P-Tyr^15^- Cdk1) at 16–24 h (Figure 5a). Furthermore, the level of phosphorylated histone H3 on Ser^10^ (P-Ser^10^-H3) was significantly increased by serum treatment, and this increase was abrogated by CompC treatment at 8–24 h. The increase in the inhibitory tyrosine phosphorylation of Cdk1 in CompC-treated cells might cause a reduction in P-Ser^10^-H3, resulting in the blockage of chromosomal condensation and segregation during mitosis.

Inhibitory tyrosine phosphorylation of Cdk1 can be controlled by other protein kinases, such as Wee1 and Myt1. In this study, we also observed that CompC treatment increased the level of phosphorylated Wee1 at Ser^642^ (P-Ser^642^-Wee1) in the absence and presence of serum during the experimental period (Figure 5a). The level of Myt1 protein was enhanced by the serum itself but not altered by CompC treatment at any time point. These results suggest that tyrosine phosphorylation of Cdk1 correlates with the serine phosphorylation status of Wee1. The levels of total Cdk1, Wee1, Myt1, and histone H3 were not significantly altered by CompC treatment.

Cdk activity can be inhibited by the binding of Cdk inhibitors, such as p21, p27, and p16, to Cdk-cyclin complexes [15]. Therefore, the levels of these Cdk inhibitors were examined by western blot analysis. Although the levels of Cdk inhibitors were markedly increased after serum treatment, the effects of CompC on Cdk inhibitors did not differ depending on the time points (data not shown). In contrast, CompC treatment increased the levels of total and phosphorylated p53 and it did not increase p21 levels at any time point (Figure 5a). The role of CompC-induced activation of p53 and reduction of Cdk inhibitors needs to be clarified.

Treatment with the Wee1 inhibitor adavosertib (MK-1775, Adavo) at specific concentrations (25–50 nM) interferes with CompC-dependent P-Tyr^15^- Cdk1 and P-Ser^15^-p53 induction, as well as CompC-dependent P-Ser^10^-H3 reduction (Figure 5b). To confirm the effect of Wee1 on CompC-induced cell viability reduction, the cell viability of 5–10 μM CompC, 50–100 nM Adavo alone, and CompC/Adavo co-treated cells was measured by MTT assay. CompC-induced cell viability reduction was recovered by co-treatment with 50 nM Adavo, compared with CompC-treated cells in the absence of Adavo (Figure 5c). However, this recovery effect was not visible at higher concentrations. Since it was observed that Adavo alone (5–500 nM) reduced the viability of Renca cells in a dose-dependent manner (Figure 5d), its recovery effect may be offset and detrimental at higher concentrations. These results suggest that CompC exerts inhibitory effect on mitosis and G2/M cell cycle arrest by inhibiting phosphorylation of Cdk1 on tyrosine through Wee1 activation and that of histone H3 subsequently.

### 2.4. CompC Reduces the Cell Adhesion, Migration and Invasion of Renca Cells

The adhesion of cancer cells to the extracellular matrix (ECM) or the cell-ECM interactions is an important step in metastasis. We examined the effect of CompC on in vitro adhesion of Renca cells to Matrigel-coated plates. The cells were seeded onto the plates and allowed to attach for 1 h in the presence of the vehicle or CompC (at 0, 2.5, 5, and 10 μM concentrations). As shown in Figure 6a,b, CompC significantly (*** *p* < 0.001) inhibited the adhesion of Renca cells to Matrigel-coated plates in a concentration-dependent manner.

We next determined whether CompC inhibited the migration and invasive behavior of Renca carcinoma cells. For the cell migration experiment, Renca cells at 50–60% confluence were serum starved by incubation with SFM for 24 h, and some areas were denuded. The cells were then incubated in FBS medium in the presence of the vehicle or various concentrations (2.5–10 μM) of CompC. Cell migration was increased by FBS, but co-treatment with FBS and CompC significantly inhibited cell migration in a dose- and time-dependent manner (Figure 7a,b). Invasion assay was performed using the ECMatrix cell invasion assay kit. Cell invasion was determined in the presence of the vehicle or CompC for 24 and 48 h. As shown in Figure 7c,d, CompC at a concentration of 2.5–10 μM markedly suppressed the invasion of Renca cells at 24 and 48 h. These results suggest that CompC exhibits anti-invasive behavior towards Renca cells.

### 2.5. CompC Inhibits Renca Tumor Growth in BALB/c Syngeneic Mice

Because CompC inhibited mitosis of Renca epithelial carcinoma cells in vitro and had an inhibitory effect on cell cycle progression, adhesion, migration, and invasion, it was assumed that CompC would inhibit in vivo Renca tumor growth in BALB/c syngeneic mice. Renca cells (1 × 10^6^) were subcutaneously injected into 6-week-old BALB/c mice. Ten days after tumor cell injection, the vehicle or CompC (2.5 mg/kg/day) was intraperitoneally injected into tumor-bearing BALB/c mice once a day for seven days. During the experimental period, body weight and tumor volume in BALB/c mice were measured daily.

The body weight did not show any significant differences between the vehicle- and CompC-injected groups (Figure 8a). However, there was a marked difference in the tumor volume between the treatment groups, four–seven days after CompC injection (χ^2^ = 27.7, df = 1, *p* < 0.001, Kruskal–Wallis nonparametric test). Post-hoc comparisons showed statistically significant differences between the vehicle and CompC treatment groups on certain days (Mann–Whitney U = 0.000, *p* < 0.01, Mann–Whitney U tests). Those were marked with an asterisk each (* *p* < 0.01). The in vivo growth rate of Renca tumors in CompC-injected mice was lower than that in the vehicle-injected mice. The average tumor volume in the CompC-injected group was reduced by more than 40% compared with that in the vehicle-injected group at the seventh day (Figure 8b). Seven days after daily vehicle or CompC injections, tumor samples from Renca-bearing mice were isolated and photographed. As shown in Figure 8c, masses excised from CompC-injected mice were smaller and bloodless compared to those excised from vehicle-injected mice. These data suggest that CompC inhibits in vivo Renca tumor growth in BALB/c syngeneic mice.

## 3. Discussion

Non-cytotoxic concentrations of CompC have been shown to inhibit the proliferation and viability of various normal as well as cancerous cells, including HDFs, MRC-5 human lung fibroblasts, BEAS-2B human bronchial epithelial cells, rat aortic vascular smooth muscle cells (VSMC), human U251, rat C6 glioma cell lines, and glioblastoma cells [12,14,18]. In the present study, we also observed that CompC reduced the viability of various renal carcinoma cell lines including Renca, A498, Caki-1, and Caki-2 (Figure 1a). CompC is known to cause G0/G1 cell cycle arrest but does not induce cell death in HDFs [14]. However, CompC produces apoptotic cells in glioma and glioblastoma cells [12,18]. In the present study, no significant cell death was observed in Renca cells (Figure 1b and Figure 2). Similarly to our study on Renca cells, Jang et al. showed that treatment with 15 μM CompC alone for 24 h did not significantly induce apoptosis (~1.5%) as determined by flow cytometry after PI staining, but CompC enhanced TRAIL-induced apoptosis in Caki cells [31]. The mechanism of action of CompC in cell proliferation and death may differ depending on the cell type. Here, the molecular mechanisms underlying the CompC-induced anti-proliferation effect were studied further in Renca cells.

In a recent study using HDFs, the biochemical effects of CompC were attributed to its inhibitory action on PDGFRβ tyrosine kinase activity [14]. This initial action on PDGFR affected all downstream signaling proteins, such as ERK1/2, Akt, PI3K, and PLCγ at early time points (1–60 min), and cell cycle regulatory proteins, such as phosphorylated pRB, cyclins, Cdks, and Cdk inhibitors at late time points (8–24 h). The inhibitory effect of CompC on PDGFR and downstream signaling proteins has also been demonstrated in other PDGFR-expressing cells, including MRC-5, BEAS-2B, VSMC, and A172 human glioblastoma cells [14]. Its inhibitory action in the initial step causes G0/G1 cell cycle arrest in HDFs [14]. However, in the present study, flow cytometric analysis revealed that CompC caused G2/M cell cycle arrest in Renca cells (Figure 2). These results agree with previous reports that CompC causes G2/M cell cycle arrest and apoptotic cell death in glioma cell lines [12,18] as well as B16F1 melanoma cells [15]. The molecular mechanism of CompC-induced G2/M cell cycle arrest in Renca cells and other tumor cells may be different from that of G0/G1 cell cycle arrest in HDFs.

Although CompC is a widely known AMPK inhibitor, some of its biological effects are AMPK independent [32,33]. A previous study reported that the inhibitory effect of CompC on PDGFR phosphorylation and downstream signaling events is AMPK independent [14]. AMPK-independent anti-proliferation and cell death effects of CompC have also been suggested by Liu et al. [12]. An additional study reported that the inhibition of hypoxia-induced HIF-1α activation by CompC is independent of AMPK [20]. However, a previous study on U251 glioma cells demonstrated that knock-down of AMPK by small interfering RNA (siRNA) targeting AMPK mimicked CompC-induced G2/M cell cycle arrest, but failed to induce oxidative stress and apoptosis [18]. These results suggest that AMPK inhibition is required for G2/M cell cycle arrest but is not sufficient for CompC-mediated apoptotic death of glioma cells [18]. It is not clear whether CompC-induced G2/M cell cycle arrest in Renca cells is AMPK-dependent. Since the selectivity of CompC for AMPK remains uncertain [33], it is postulated that it may induce other pathways for cell cycle arrest; therefore, further studies are required.

In this study, it was shown that PDGF- and serum-induced phosphorylation/activation of PDGFR as well as downstream signaling proteins, including Akt, PI3K, and PLCγ, were reduced by CompC treatment in Renca cells (Figure 3a,b). Its inhibitory effects on PDGFR-dependent signaling were in agreement with a previous report which showed that CompC inhibits PDGFR along with downstream signaling proteins in HDFs and other PDGFR-expressing cells [14]. Unlike these signaling proteins, the levels of phosphorylated ERK1/2 did not agree with those in previous reports. In Renca cells, CompC treatment increased ERK1/2 phosphorylation (Figure 3b). ERK1/2 serine/threonine protein kinases that participate in the Ras-Raf-MEK-ERK signal transduction cascade. Recent evidence indicates that ERK signaling can function as a tumor suppressor as well as a more common pro-oncogenic signal, presumably via the regulation of various processes, including cell cycle progression, proliferation, survival, adhesion, invasion, differentiation, and apoptosis [34,35,36]. The predominant effect depends on the intensity of the signal and the context or tissue in which the signal is aberrantly activated [36,37]. Since ERK1/2 proteins are associated with G1 cell cycle progression, it is possible that CompC-induced activation of ERK1/2 plays a key role in overcoming CompC-induced cell cycle arrest in the G1 phase. In addition, cisplatin-induced ERK activation contributes to p53 accumulation and elevated levels of Ser^15^-phosphorylated p53 [38]. CompC-induced ERK1/2 signaling may result in an increase in phosphorylated p53 (P-Ser^15^-p53) in Renca cells (Figure 5a).

Previously, the involvement of ROS in ERK1/2 activation has been demonstrated [15,26,39]. ROS-induced ERK activation has also been implicated in cell cycle arrest [27,28]. Furthermore, it has been reported that the pro-apoptotic action of CompC involves induction of oxidative stress [18]. In our study as well, we observed that CompC increased the level of ROS (Figure 4a,b) and phosphorylated forms of ERK1/2 in Renca cells (Figure 3b), while the antioxidant NAC reduced CompC-induced ERK activation (Figure 4c) and G2/M cell cycle arrest (Figure 4d,e), enhancing cell viability (Figure 4f). These results suggest that CompC-induced ROS production and ROS-dependent ERK1/2 activation may play a partial role in CompC-induced anti-proliferation in Renca cells. The mechanisms by which ROS activates ERK1/2 kinases are unclear; however, the oxidation of upstream kinases and inactivation or degradation of MAPK phosphatases may provide a plausible explanation [26]. Based on the negative crosstalk between AMPK and ERK signaling [40], it is possible that CompC activates ERK signaling via AMPK inhibition. The interrelationship between AMPK inhibition, ROS production, and ERK1/2 activation remain to be clarified further.

The entry of eukaryotic cells into mitosis is triggered by Cdk1 activation, whose inactivation mediates rapid arrest in the G2/M phase in response to DNA damage [29,30]. The activity of Cdk1 is controlled by several steps, including cyclin (A or B) binding, phosphorylation of Cdk1 at Thr^161^ by a CDK-activating kinase (CAK), and dephosphorylation of Cdk1 at Thr^14^ and Tyr^15^ by Cdc25 phosphatase [29,30,41]. In addition, activated Cdk1 can phosphorylate Cdc25 at multiple sites, further activating Cdc25 to form a positive feedback loop [42].

In contrast, the inhibitory phosphorylation of Cdk1 kinase at Tyr^15^ and Thr^14^ is carried out by Wee1 and Myt1 protein kinases, respectively [41,43]. Furthermore, it is generally known that a portion of Wee1 is phosphorylated at Ser^642^ during the interphase of the cell cycle, and Wee1 inactivation may be accompanied by dephosphorylation of P-Ser^642^ [44]. Phosphorylation of histone H3 by active Cdk1 kinase is involved in nucleosome structure and mitosis progression [45]. The present study revealed that the levels of Cdk1 and Tyr^15^-phosphorylated Cdk1, and Ser^642^-phosphorylated Wee1 were induced by CompC treatment at later time points (8–24 h) in Renca cells (Figure 5a). The phosphorylation status of Cdk1 at Tyr^15^ and Wee1 at Ser^642^ correlated well. These findings suggest that activation of Wee1 might enhance the Tyr^15^-phosphorylation and inactivation of Cdk1, resulting in reduced phosphorylation of histone H3 (Figure 5a), G2/M cell cycle arrest (Figure 2), and reduced cell viability (Figure 1) in CompC-treated Renca cells.

The present study indicates that CompC induces G2/M arrest via activation of Wee1. Treatment with Adavo, a small molecule Wee1 inhibitor, resulted in partial recovery of CompC-induced cell viability reduction only at a certain concentration (Figure 5c). At higher concentrations, Adavo inhibited cell viability in a dose-dependent manner (Figure 5c), showing no recovery of CompC-dependent effects on cell viability. In contrast, it enhanced the cytotoxic effect of CompC in Renca cells (Figure 5c). Previously, the combination of Adavo with various DNA-damaging agents, including 5-fluorouracil, cisplatin, and radiation, has been reported to produce robust anti-tumor activity [46,47]. Adavo enhances the cytotoxic effects of 5-fluorouracil in p53-deficient human colon cancer cells, presumably by premature entry into mitosis via inhibition of Cdk1 Tyr^15^ phosphorylation, induction of histone H3 phosphorylation, abrogation of DNA damage checkpoints, and G2 block [46]. Wee1 inhibition by Adavo is especially effective in *TP53/RAS*-mutant metastatic colorectal cancer [48].

AMPK activation has been shown to be associated with cell cycle arrest through the activation of p53 and the subsequent induction of Cdk inhibitor p21 [15,49]. Although CompC is an inhibitor of AMPK activity, it increased the level and activity of p53 but did not increase the level of p21 at 8–24 h (Figure 5a). Cdk1 has been shown to bind with p53 and induce down-regulation of Cdk1 kinase activity [50,51]. The increased level of p53 by CompC might be responsible in part for inactivation of Cdk1, reduced phosphorylation of histone H3, and G2/M cell cycle arrest in CompC-treated cells.

The adhesion of cancer cells to the ECM and their interactions are critical steps in metastatic and invasive processes [23,52]. Furthermore, tumor invasion are common features to be specially considered for the evaluation of cancer progression and metastasis, which may result in high morbidity and mortality [53]. A previous study reported that CompC reduced the anti-invasive effect induced by fenofibrate, a peroxisome proliferator-activated receptor alpha (PPARα) agonist and a lipid-lowering agent, in human oral cancer CAL 27 cells in an AMPK-dependent manner [24]. We also observed that the CompC treatment markedly inhibited cell adhesion onto Matrigel-coated plates (Figure 6) and the invasive/migratory abilities of Renca cells (Figure 7). In addition, we showed that CompC significantly suppressed tumor growth in the Renca xenografts of syngeneic BALB/c mice (Figure 8).

Taken together, our results suggest that CompC may exert anti-tumor effect in vivo by suppressing cell cycle progression, adhesion, migration, and invasion. As shown in the schematic representation (Figure 9), CompC suppresses cell cycle progression in Renca cells by inhibiting PDGFR-dependent signal transduction including Akt and inducing ROS-dependent cell cycle arrest through ERK1/2 activation, Wee1 activation, Cdk1 inhibition, and subsequent histone H3 inactivation. Activation of p53 through ROS production or other signaling pathways also plays an important role in cell cycle arrest. In addition to inhibiting cell cycle progression or cell cycle arrest, CompC also inhibits Renca cell adhesion, migration, and invasion. The molecular mechanisms for CompC-dependent suppression of adhesion, migration, and invasion have yet to be clarified.

## 4. Materials and Methods

### 4.1. Materials

The 3-(4,5-Dimethylthiazol-2-yl)-2,5-diphenyltetrazolium bromide (MTT), 2′,7′-dichlorodihydrofluorescein diacetate (DCFH-DA), dimethyl sulfoxide (DMSO), mouse anti-β-actin monoclonal antibody (#A5441), propidium iodide (PI), crystal violet, and RNase A were obtained from Sigma-Aldrich (St. Louis, MO, USA). CompC was purchased from Calbiochem (San Diego, CA, USA). Dulbecco’s modified Eagle’s medium (DMEM) and RPMI medium were purchased from Corning (Corning, NY, USA). Fetal bovine serum (FBS), penicillin, and streptomycin were purchased from Gibco/BRL Life Technologies, Inc. (Carlsbad, CA, USA). Antibodies against Akt (#9272), phosphorylated Akt on Ser^473^ (P-Ser^473^-Akt) (#4051), ERK1/2 (#9102), phosphorylated ERK1/2 on Thr^202^/Tyr^204^ (P-ERK1/2) (#4370: rabbit mAb), AMPKα (#2532), phosphorylated AMPKα on Thr^172^ (P- Thr^172^-AMPKα) (#2535), PI3K p85 (#4292), phosphorylated PI3K p85 on Tyr^458^ and p55 on Tyr^199^ (P-PI3K) (#4228), PLCγ1 (#2822), phosphorylated PLCγ1 on Tyr^783^ (P-Tyr^783^-PLCγ1) (#2821), Cdk1 (#9112), phosphorylated Cdk1 on Tyr^15^ (P-Tyr^15^-Cdk1) (#9111) and on Thr^161^ (P-Thr^161^-Cdk1) (#9114), cyclin B1 (#4138), phosphorylated Wee1 on Ser^642^ (P-Ser^642^-Wee1) (#4910), Myt1 (#4282), p53 (#9282), P-Ser^15^-p53 (#9284), histone H3 (#9715), and phosphorylated histone H3 on Ser^10^ (P-Ser^10^-H3) (#3377) were purchased from Cell Signaling Technology (Danvers, MA, USA). Antibodies against cyclin A (#sc-751), Wee1 (#sc-325), p21 (#sc-397), and PLCγ1 (#sc-166938) were purchased from Santa Cruz Biotechnology (Dallas, TX, USA). Horseradish peroxidase (HRP)-conjugated anti-rabbit and anti-mouse secondary antibodies were from Vector Laboratories (Burlingame, CA, USA); the protein assay kit from Bio-Rad Laboratories (Hercules, CA, USA), Matrigel^TM^ Basement Membrane Matrix from BD Biosciences (Sparks, MA, USA). Adavosertib was purchased from MedChemExpress (Monmouth Junction, NJ, USA). the SPLInsert^TM^ Matrigel Transwell invasion assay kit was obtained from SPL life science (Pocheon, Korea).

### 4.2. Cell Culture

*Mus musculus* renal adenocarcinoma Renca cells were purchased from American Type Culture Collection (Manassas, VA, USA) and cultured in DMEM with high glucose (4.5 g/L) supplemented with 10% (*v*/*v*) FBS, 100 U/mL penicillin and 100 μg/mL streptomycin. Other renal cancer cell lines (RCCs: A-498, Caki-1, and Caki-2) were obtained from the Korean Cell Bank (Seoul, Korea). RCC cells were cultured in DMEM medium. All RCCs were maintained in a humidified 5% (*v*/*v*) CO_2_ incubator at 37 °C for treatment, cells were cultured for one day, and incubated with the vehicle or various concentrations of CompC for the indicated times

### 4.3. MTT Assay

Renca cells were seeded in quadruplicate into 24-well plates at a density of 1 × 10^4^ cells/well in 1 mL of culture medium, incubated for one day, and treated with the vehicle or various concentrations of CompC. After incubation at 37 °C for two days, the cells were washed once with ice-cold DMEM, then 500 μL of MTT solution (0.5 mg/mL in DMEM) was added, followed by incubation for 2 h at 37 °C in the dark. After removing the MTT solution, 200 μL of DMSO was added to dissolve the blue formazan in living cells. Absorbance was read at 570 nm using an ELISA reader (VICTOR3, PerkinElmer Life and Analytical Sciences, Turku, Finland). The experiments were repeated three times.

### 4.4. Flow Cytometry

The DNA content in the phases of the cell cycle (SubG1, G1, S, G2/M) was determined by flow cytometry after staining the ethanol-fixed cells with PI, as described previously [15]. The cells (5 × 10^4^ cells/4 mL/well) were seeded onto 6-well plates. After one day of culture, the medium was replaced with fresh SFM and incubated overnight. The cells were then treated with the vehicle or 10 μM CompC. After incubation for the indicated times (8, 16, and 24 h) at 37 °C, the cells were harvested, washed twice with PBS, and fixed in ice-cold 70% ethanol overnight at 4 °C. For PI staining, fixed cells were pelleted by centrifugation at 500× *g* at 4 °C for 5 min and washed twice with ice-cold PBS. After suspending cells in 1 mL PBS, DNase-free RNase A (0.1 mg/mL) and PI (20 μg/mL) were added. Aggregated cells were removed by transfer to FACS tubes through a cap filter. The cell suspension was incubated for 30 min at 37 °C in the dark. The stained cells were kept on ice in the dark until analysis with a FACSCalibur flow cytometer (Becton Dickinson, San Jose, CA, USA), with excitation at 488 nm and emission at 605–617 nm (FL-2 channel).

### 4.5. Western Blot

Protein levels and phosphorylation status were examined by western blot analysis, as described previously [14]. Renca cells were serum-starved for 16 h, treated with 10% FBS in the absence or presence of 10 μM CompC, washed twice with ice-cold PBS, and lysed for 20 min in a lysis buffer (50 mM Tris–HCl, pH 7.5, 150 mM NaCl, 2 mM EDTA, 1 mM EGTA, 1 mM Na_3_VO_4_, 10 mM NaF, 1 mM DTT, 1 mM PMSF, 25 μg/mL leupeptin, 25 μg/mL aprotinin, 5 mM benzamidine, and 1% Ingepal CA630). The cells and lysates were collected in 1.5 mL Eppendorf tubes, sonicated for 10 s on ice, and rocked at 4 °C for 15 min. The cell lysates were centrifuged at 14,000 g for 15 min and the supernatants were collected. Protein concentrations of the lysates were determined using a Bio-Rad protein assay kit, as described by the manufacturer. Cell lysates were mixed with Laemmli sample buffer and boiled for 5 min. The proteins (30 μg/lane) were resolved on 8–12% SDS-polyacrylamide gels and transferred to Immobilon PVDF membranes. The blots were blocked with a solution containing 5% nonfat dried milk or 5% BSA and 0.1% Tween 20, and treated with the appropriate antibodies in the blocking solution overnight. The blots were then washed and probed further with HRP-conjugated goat anti-rabbit or anti-mouse secondary IgG (1:5000). Immune complexes were visualized using an ECL detection system, as described by the manufacturer.

### 4.6. Measurement of Intracellular ROS

The level of intracellular ROS was evaluated by flow cytometry after incubating the cells with the redox-sensitive dye, DCFH-DA. Cells were seeded onto 6-well plates at a density of 1 × 10^5^ cells/well in 4 mL culture medium and incubated for one day. Cells were then treated with vehicle or CompC (10 μM) in SFM or in the culture medium containing 10% FBS for 1 and 6 h. For the inhibition of ROS generation, cells were pre-treated with 5 mM of the antioxidant N-acetyl cysteine (NAC) for 1 h before CompC treatment. The cells were washed once with 1 X PBS, and a culture medium containing 10 μM DCFH-DA was added to each well. After incubation for 30 min at 37 °C, DCF-labeled cells were washed once with 1X PBS, trypsinized with 1X trypsin-EDTA for 1–2 min, washed twice with ice-cold PBS, and then suspended in 1X PBS. The fluorescence intensities of the labeled cells were determined using a FACSCalibur flow cytometer at excitation and emission wavelengths of 488 and 525 nm, respectively. Ten thousand events were analyzed per sample.

### 4.7. Matrigel Adhesion Assay

Renca cells suspended in SFM were seeded at 2 × 10^4^ cells/well in Matrigel-coated 96-well plates. The cells were treated with the vehicle or 10 μM CompC and allowed to attach during incubation for 1 h at 37 °C. After shaking the plate for 15 s on a 2000 rpm rocker, the medium was aspirated, and non-adherent cells were removed by washing three times with PBS at room temperature (RT). Adherent cells were fixed by incubation with 96% ethanol for 15 min at RT, washed twice with PBS, and then stained with 0.1% crystal violet for 30 min at RT. After washing with tap water, stained cells were air-dried for 24 h and photographed using an inverted microscope. The percentage of cell adhesion was calculated by extracting the dye with SDS-ethanol solution (5% SDS, 50% ethanol) and measuring the absorbance at 570 nm.

### 4.8. Cell Migration Assay

Renca cell migration was assessed based on the ability of cells to migrate into an acellular area [15]. The cells were seeded at 1 × 10^5^ cells/well onto a 12-well plate and grown in culture medium for one day. After incubation for one day in SFM to induce quiescence, a few areas were denuded using a sterile yellow pipette tip (1 mm wide), and the cells were then incubated in FBS culture medium with the vehicle or various concentrations of CompC for two days. Photographs of Renca cells were taken at the indicated times using an inverted light microscope, and the number of Renca cells that migrated into the acellular area was counted using ImageJ software. The percentage of wound closure was calculated and plotted.

### 4.9. Cell Invasion Assay

Cell invasion was assayed using the SPLInsert^TM^ Matrigel Transwell invasion assay kit that includes polycarbonate membranes with 8 μm pores and a layer of reconstituted basement membrane matrix, according to the manufacturer’s instructions. Each lower chamber was filled with 500 μL medium containing 10% FBS as the chemo-attractant in the presence of vehicle or 10 μM CompC. Renca cells (1 × 10^5^) were suspended in 300 μL serum-free medium and carefully transferred to the upper chamber of the device. The chamber was incubated for 24 h at 37 °C in a humidified atmosphere containing 5% CO_2_. The non-invading cells were then completely removed from the upper chamber with a cotton-tipped swab. Cells that had passed through the polycarbonate membrane and invaded into the lower surface were fixed in 75% methanol for 15 min, stained with crystal violet, and photographed using a light microscope. The stained cells were quantified using the ImageJ software (NIH, Bethesda, MD, USA).

### 4.10. In Vivo Xenograft Experiment

BALB/c mice (age: 6 weeks, male) were purchased from Orient Bio Inc. (Gapyung, Korea) and housed (4 per cage) in a chamber with controlled temperature and humidity. The mice were maintained on a standard diet with food and water provided *ad libitum* in an animal facility accredited by the Korean Association for the Accreditation of Laboratory Animal Care. The animal protocol was reviewed according to the Institutional Animal Care and Use Committee protocol, and was approved by the Committee for Care and Use of Laboratory Animals at Kyung Hee University (KHSASP-19-416). Before the in vivo xenograft experiment, the mice were anesthetized by methoxyflurane (isofuran) inhalation, and their backs were shaved. Renca cells (1 × 10^6^ in 100 μL PBS) were subcutaneously injected into the right side of the backs of BALB/c mice. Ten days after Renca injection, vehicle or CompC was dissolved in isotonic saline and 100 µL aliquots (corresponding to 2.5 mg/kg/day) were intraperitoneally injected into the tumor-bearing BALB/c mice. During the experimental period, the weight and tumor volume of the Renca cells in BALB/c mice were examined. Seven days after daily vehicle/CompC injection, the animals were anesthetized with a mixture of rumpun and zoletil (1:5), 8 mice per group were euthanized using a CO_2_ chamber, and tumors were excised and photographed.

### 4.11. Statistical Analysis

All data are presented as the mean ± SD of at least three repeat experiments. Statistical comparisons for in vitro experiments were performed using one-way analysis of variance (ANOVA), followed by Dunnett’s T3 post-hoc test (SPSS, IBM, Seoul, Korea) to assess significant differences between the groups. Kruskal–Wallis H tests for non-parametric statistics and post-hoc comparisons with Bonferroni-corrected Mann–Whitney U tests (SPSS; IBM, Seoul, Korea) were also performed to assess treatment effects on daily tumor growth in the Renca-bearing BALB/c syngeneic mouse model. *p* values were considered statistically significant at *p* < 0.05.

## 5. Conclusions

CompC inhibited the growth of Renca xenografts in a syngeneic BALB/c mouse model. Inhibition of in vivo tumor growth may be attributed to the anti-proliferative effect of CompC via G2/M cell cycle arrest, and the blocking of tumor cell adhesion and invasion. These findings suggest that CompC may have a potential therapeutic use in the treatment of renal cell carcinoma. 

## Figures and Tables

**Figure 1 ijms-23-09675-f001:**
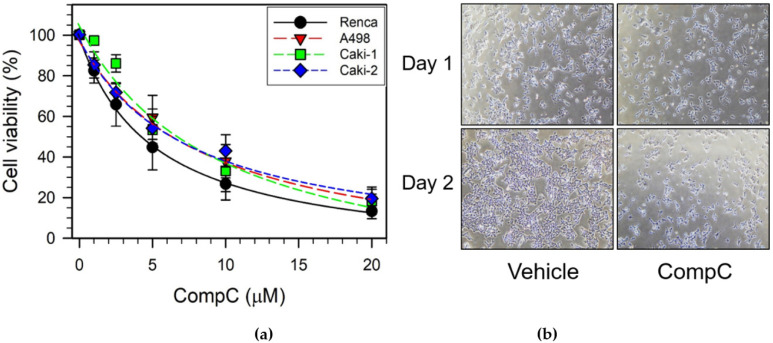
Compound C (CompC) reduced the viability of Renca renal carcinoma cells in a dose-dependent manner. (**a**) Renca, A498, Caki-1, and Caki-2 cells were seeded onto 24-well plates, cultured for one day, and treated with the vehicle, dimethyl sulfoxide (DMSO), or various concentrations of CompC (0, 1, 2.5, 5, 10, and 20 μM) in the culture media. After two days, a 3-(4,5-dimethylthiazol-2-yl)-2,5-diphenyltetrazolium bromide (MTT) assay was performed and the percentage of cell viability is plotted as the mean ± standard deviation (SD) with n = 20. (**b**) Subconfluent Renca cells were cultured in the presence of vehicle or 10 μM CompC for one–two days. The cell morphology was observed through and photographed by an inverted microscope (100×).

**Figure 2 ijms-23-09675-f002:**
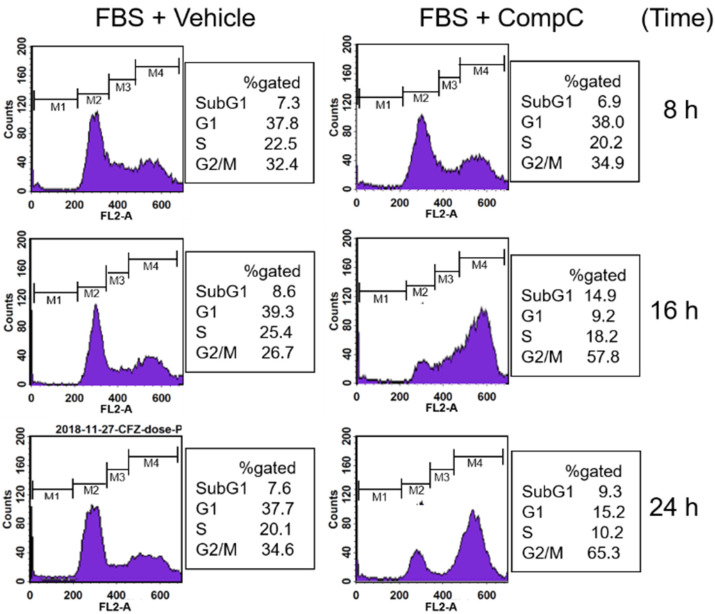
CompC treatment resulted in G2/M cell cycle arrest. Renca cells were seeded onto 6-well plates, cultured for one day and serum-depleted by incubation with serum free medium (SFM) overnight. Cells were pre-treated with the vehicle or 10 μM CompC for 1 h and then stimulated with 10% fetal bovine serum (FBS) for the indicated times. The DNA content was evaluated by flow cytometry after staining the nuclei with propidium idodide (PI). The percentage of cells in each phase is shown on the right.

**Figure 3 ijms-23-09675-f003:**
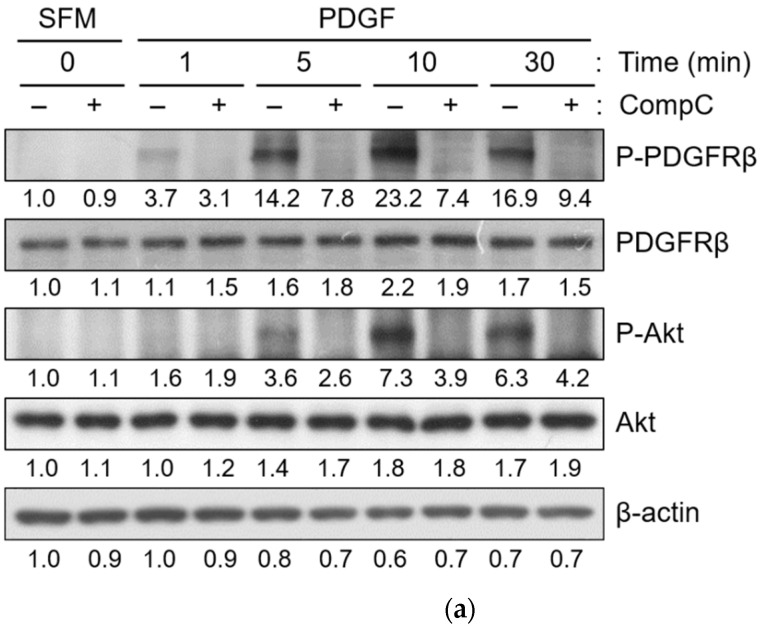
CompC reduced platelet-derived growth factor (PDGF) and serum (FBS)-induced activation/phosphorylation of PDGF receptor (PDGFR) as well as the downstream signaling proteins. Renca cells were cultured for one day and then serum-starved by incubation with SFM overnight. Cells were pre-treated with the vehicle (−) or 10 μM CompC for 1 h and then treated with a medium containing PDGF (**a**) or 10% FBS (**b**) for the indicated times. The cells were lysed, and the total cell extract (30 μg per lane) was examined by western blot analysis using antibodies against phospho-PDGFRβ (P-PDGFRβ), PDGFRβ, P-Akt, Akt, P-ERK1/2, ERK1/2, P-PI3K, PI3K, P-PLCγ, and PLCγ, with β-actin as internal control. The band densities were normalized against β-actin and the fold changes compared to that of control (SFM/0 min/−CompC) are written under each band.

**Figure 4 ijms-23-09675-f004:**
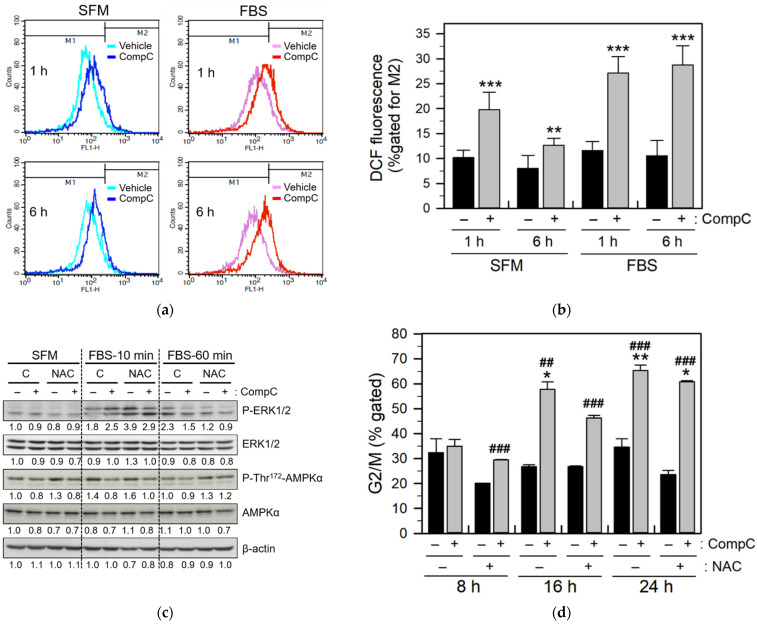
CompC generated reactive oxygen species (ROS), which induced G2/M cell cycle arrest and cell viability reduction. (**a**,**b**) Renca cells were treated with the vehicle (−) or 10 μM CompC in the SFM or FBS medium for 1 and 6 h, and the levels of intracellular ROS were evaluated by flow cytometry after incubating cells with DCF-DA for 30 min (**a**). The percentage of cells gated in the M2, is plotted as the mean ± SD (**b**). (**c**–**e**) Renca cells were pretreated with PBS (C) or 5 mM N-acetyl cysteine (NAC), an ROS scavenger, for 1 h before vehicle or CompC treatment. Cells were stimulated with 10% FBS for the indicated times and cell lysates were analyzed by western blot to determine ERK1/2 activation and the status of AMP-activated protein kinase (AMPK) activity using antibodies against P-ERK1/2, ERK1/2, P-Thr^172^-AMPKα, AMPKα, and β-actin as an internal control (**c**). The band densities were normalized against β-actin and the fold changes compared to that of the control (SFM/C/−CompC) are written under each band. Cell cycle analysis was also performed by flow cytometry after PI staining of cells pre-treated with vehicle, NAC and/or CompC, and stimulated with 10% FBS (**d**,**e**). The percentage of cells gated in the G2/M phase in the indicated times (**d**) and the percentage of cells gated in the subG1, G1, S, and G2/M phase at 16 h (**e**) are plotted as the mean ± SD (n = 8). (**f**) The cell viability of NAC-treated cells was measured by MTT assay and the percentage of cell viability is plotted as the means ± SD with n = 8. * *p* < 0.05, ** *p* < 0.01 and *** *p* < 0.001, compared with vehicle-treated control cells (−CompC). ^#^ *p* < 0.05, ^##^ *p* < 0.01, ^###^ *p* < 0.001, compared with vehicle- or CompC-treated cells in the absence of NAC (−NAC).

**Figure 5 ijms-23-09675-f005:**
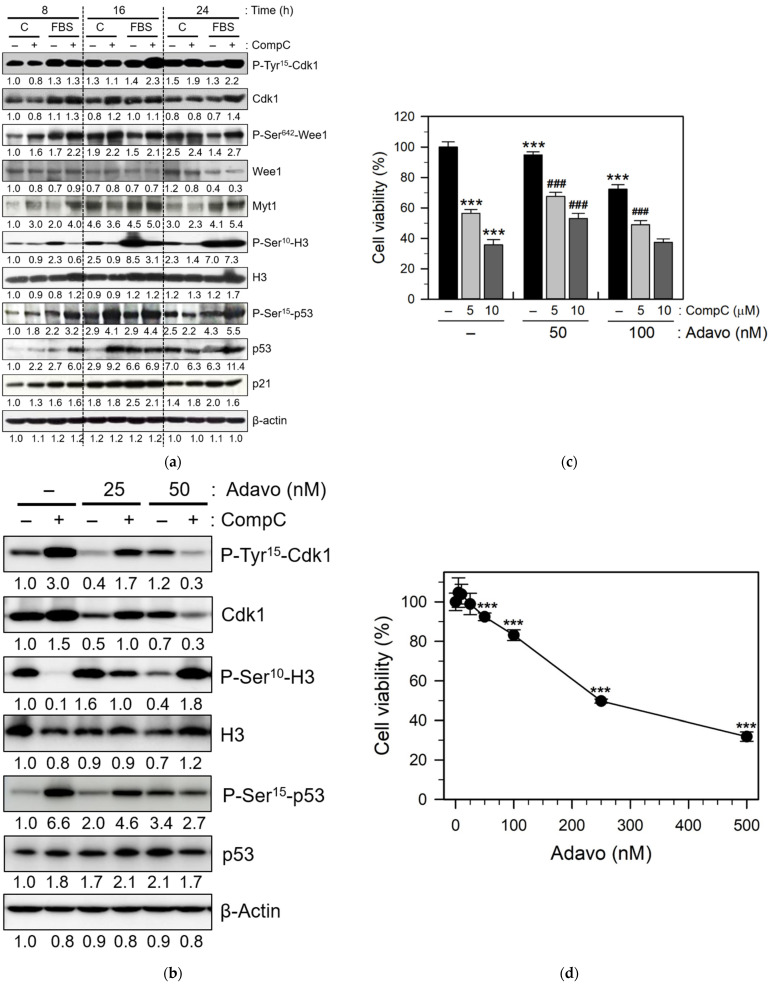
CompC increased the phosphorylation of Cdk1 on Tyr^15^ and Wee1 on Ser^642^. (**a**) Renca cells were cultured for one day and then serum-starved by incubation with SFM overnight. The cells were treated with SFM (C) or 10% FBS for the indicated times (8–24 h) in the absence (−) or presence (+) of 10 μM CompC. Cell lysates were analyzed by western blot using antibodies against phosphorylated Cdk1 (P-Tyr^15^-Cdk1), Cdk1, phosphorylated Wee1 (P-Ser^642^-Wee1), Wee1, Myt1, phosphorylated histone H3 (P-Ser^10^-H3), histone H3 (H3), phosphorylated p53 (P-Ser^15^-p53), p53, p21, and β-actin. (**b**) Renca cells treated with vehicle or 25–50 nM Adavo in the absence (−) or presence (+) of 10 μM CompC for 16 h. Cell lysates were analyzed by western blot using antibodies against P-Tyr^15^-Cdk1, Cdk1, P-Ser^10^-H3, H3, P-Ser^15^-p53, p53, and β-actin. The band densities were normalized against β-actin and the fold changes compared to that of control (vehicle/−CompC/−Adavo) are written under each band in (**a**,**b**). (**c**) The cell viability of 5–10 μM CompC-, 50–100 nM Adavo-, and CompC/Adavo-cotreated cells was measured by MTT assay, and the percentage of cell viability was plotted as the mean ± SD with n = 8. The *p*-values were determined as *** *p* < 0.001, compared with vehicle-treated control cells (−CompC/−Adavo) and ^###^ *p* < 0.001, compared to vehicle- or CompC-treated cells in the absence of Adavo. (**d**) Renca cells were cultured for one day and treated with the vehicle or 5–500 nM Adavo for two days. The cell viability was measured by the MTT assay and the percentage was plotted as the mean ± SD with n = 8. *** *p* < 0.001, compared with vehicle-treated control cells.

**Figure 6 ijms-23-09675-f006:**
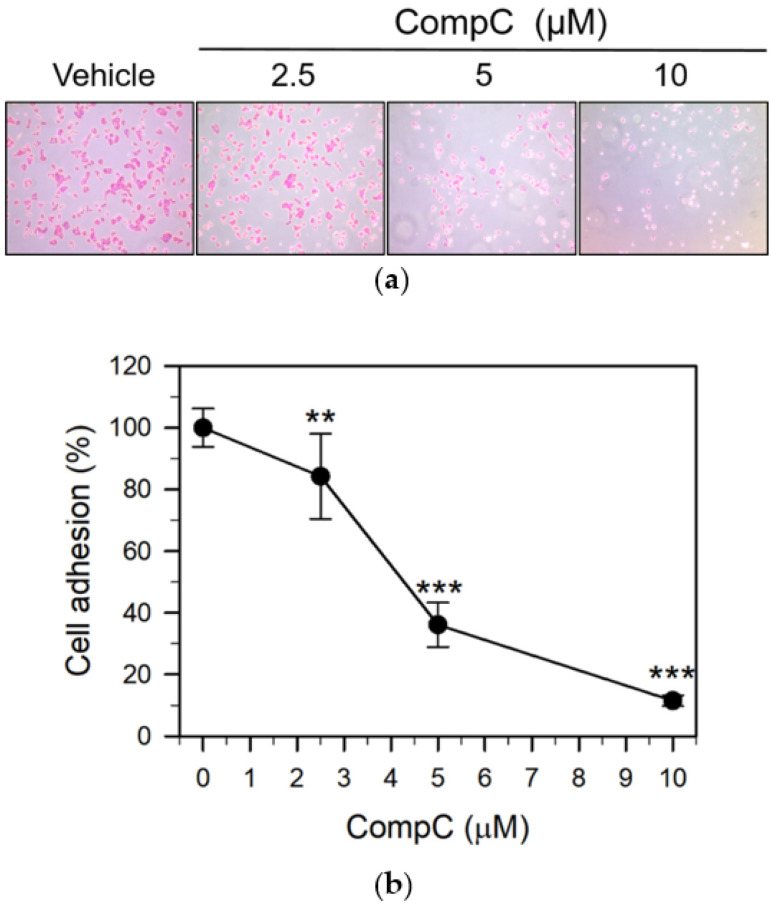
CompC inhibited the in vitro adhesion of Renca cells. A Renca cell suspension in SFM (2 × 10^4^ cells per well) was seeded to Matrigel-coated 96-well plates. The cells were treated with the vehicle or various concentrations (2.5, 5, and 10 μM) of CompC and allowed to attach during incubation at 37 °C for 1 h. Adherent cells were fixed and stained with 0.1% crystal violet for 30 min. (**a**) The adherent cells were air-dried and photographed using an inverted microscope (100×). (**b**) The percentage of cell adhesion was calculated by extracting the dye with SDS-ethanol solution (5% SDS, 50% ethanol) and measuring the absorbance at 570 nm. The percentage was plotted as the mean ± SD with n = 10. ** *p* < 0.01 and *** *p* < 0.001 compared with vehicle-treated control cells.

**Figure 7 ijms-23-09675-f007:**
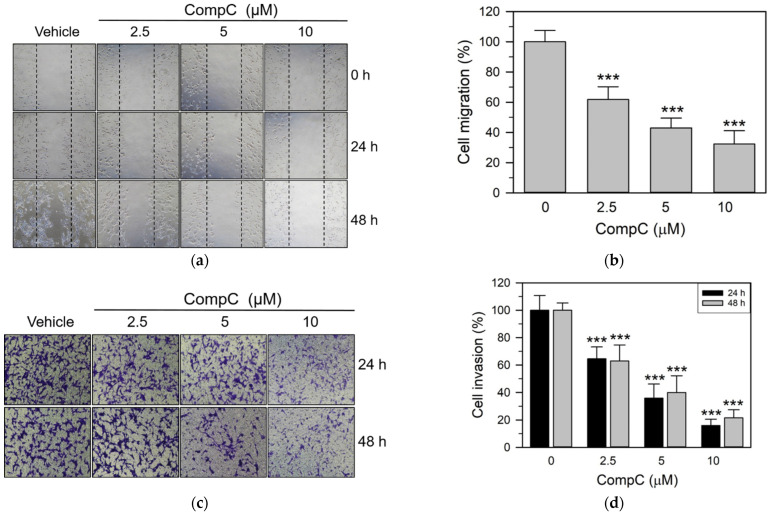
CompC inhibited the migration and invasion of Renca cells. (**a**,**b**) Cell migration was assessed based on the wound healing assay that counts the number of cells to move into an acellular area. Cells (1 × 10^5^) seeded onto a 6-cm dish and grown in culture media were serum-starved for one day in SFM and denuded in a few areas. Cells were then incubated for 24–48 h in FBS culture medium with the vehicle or various concentrations of CompC. Photographs of Renca cells were taken at 24 h and 48 h (**a**), while the number of Renca cells that had migrated into the acellular area was counted and plotted in (**b**). (**c**) Cell invasion was assayed using the SPLInsert^TM^ Matrigel Transwell invasion assay kit. Cells (1 × 10^5^) were suspended in the SFM medium containing DMSO or 2.5–10 μM CompC and carefully transferred into the upper chamber of the devices. Cells that had invaded the polycarbonate membrane to the lower surface for 24–48 h at 37 °C were fixed, stained with crystal violet, and photographed by a light microscope (100×). (**d**) The stained cells were quantified using the ImageJ software. Three independent experimental results were presented as the percentage of cell invasion relative to the vehicle-treated control. *** *p* < 0.001, compared with the vehicle-treated control.

**Figure 8 ijms-23-09675-f008:**
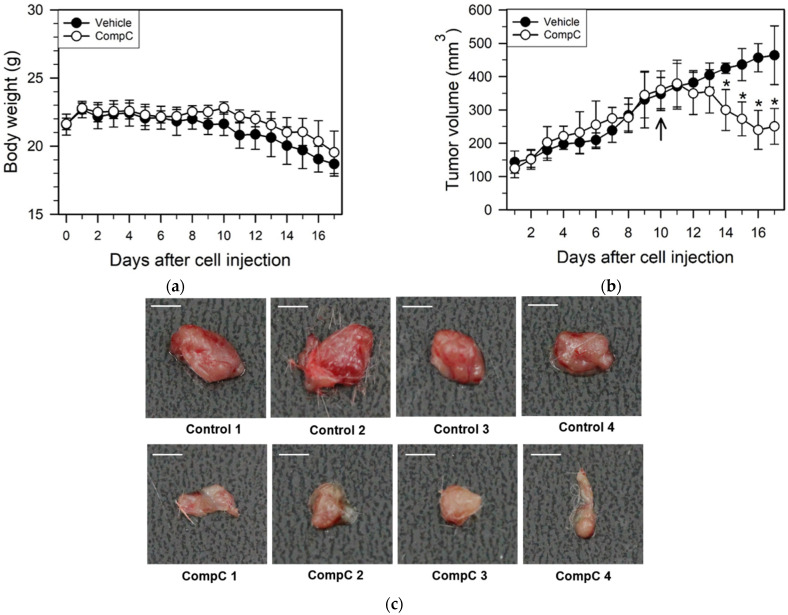
CompC inhibited the growth of Renca xenografts in BALB/c syngeneic mice. Ten days after the subcutaneous injection of Renca cells (arrow), vehicle or CompC (2.5 mg/kg/day) was intraperitoneally injected to the tumor-bearing BALB/c mice. During the experimental period, the body weight and tumor volume were recorded every day. The Kruskal–Wallis H test followed by a Bonferroni-corrected Mann–Whitney U test were used as a post-hoc comparison of the daily tumor volume in the mouse model. Significant difference in the tumor volume between vehicle and CompC injection groups at each day is marked with an asterisk. The means and SD are plotted (n = 5 for biological replicates at each time points in (**a**,**b**)). * *p* < 0.01, compared with the vehicle-injected control mice. Seven days after daily vehicle/CompC injection, the animals were euthanized, and tumors were excised and photographed (**c**).

**Figure 9 ijms-23-09675-f009:**
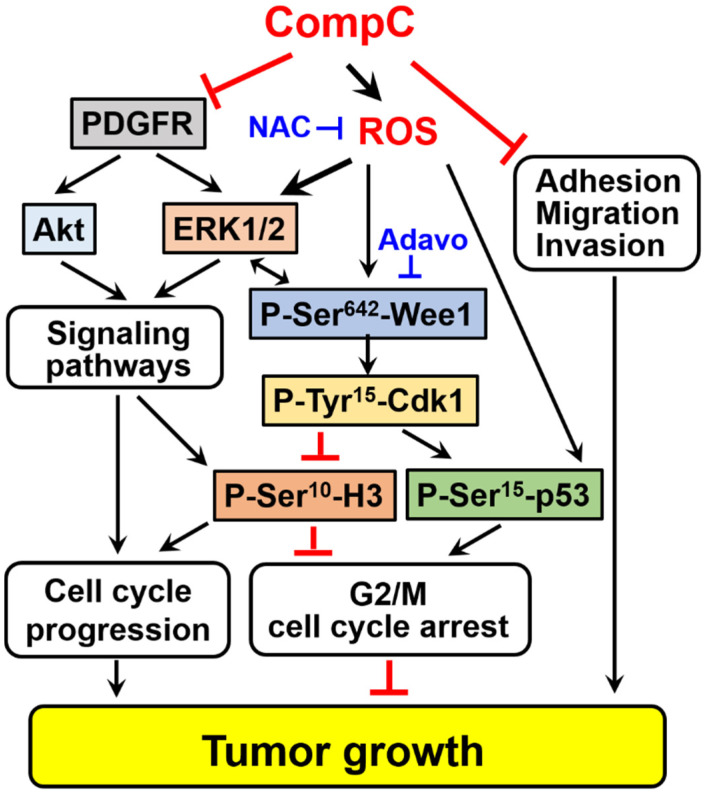
Schematic representation of the effect of CompC on tumor growth in Renca syngeneic mouse model. Arrow (→) and truncated line (┬) indicate activation and inhibition, respectively. Regulatory pathways of cell cycle progression, G2/M cell cycle arrest, adhesion, migration, and invasion diverge into tumor growth in the Renca syngeneic mouse model. The left part of the schematic indicates that CompC suppresses cell cycle progression by inhibiting PDGFR-dependent signaling pathways, including Akt. At the center of the schematic, ROS-dependent ERK1/2 activation also appears to play a role in G2/M cell cycle arrest. The ROS scavenger, N-acetyl cysteine (NAC), interferes with CompC-induced ERK1/2 activation and G2/M cell cycle arrest. ROS may also induce phosphorylation of Wee1 at Ser^642^ via ERK1/2 or other signaling pathways. Active Wee1 inactivates Cdk1 by increasing phosphorylation of Cdk1 at Tyr^15^ (P-Tyr^15^-Cdk1), which in turn results in the reduction of phosphorylated histone H3 at Ser^10^ (P-Ser^10^-H3). CompC-induced inactivation of Cdk1 and reduction of P-Ser^10^-H3 may play a role in G2/M cell cycle arrest. Activation of p53 via ROS production or other signaling pathways also plays a role in G2/M cell cycle arrest. The Wee1 inhibitor Adavosertib (Adavo) prevents the CompC-dependent increase in P-Tyr^15^-Cdk1 and P-Ser^15^-p53 levels and the CompC-dependent decrease in P-Ser^10^-H3 levels, thereby reducing G2/M cell cycle arrest. As shown in the right part of the figure, CompC inhibits Renca cell adhesion, migration, and invasion. Although the molecular mechanism remains to be clarified, CompC-dependent inhibition of adhesion, migration, and invasion may play a role in reducing tumor growth in a Renca syngeneic mouse model.

## Data Availability

The original western blot data supporting Figure 3a,b, Figure 4c and Figure 5a,b are presented.

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
