# Peer review of "Compound C Inhibits Renca Renal Epithelial Carcinoma Growth in Syngeneic Mouse Models by Blocking Cell Cycle Progression, Adhesion and Invasion"

_ijms, 2022, doi:10.3390/ijms23179675_

Round 1

Reviewer 1 Report

There is a report indicating the Compound C effects on renal cancer cells. The study has shown that Compound C led to cell growth inhibition, ROS induction, and TRAIL-induced apoptosis in the renal cancer cells (PMID: 20451517).

1.   In Figure 1 and the discussion, the author stated that Compound C did not induce apoptosis in Renca cells. However, the results are insufficient to make a conclusion based on micrographs alone. Is there any experiment to measure apoptosis with Annexin V/ PI staining? If not, the author should add an experiment showing apoptosis.

2.   It was mentioned that Renca cells and other RCC cell lines after treatment with Compound C were all measured under a microscope. Didn't other RCCs (Caki-1, Caki-2, and A498) have an apoptotic effect?

3.   The sentence explaining Figure 5b does not look correct. From the Western results, it appears that Adavosertib interferes with the reduction of P-Ser10-histone H3 and total histone H3 by compound C. It seems that the sentence arrangement of P-Tyr15-Cdk1 and P-Ser10-histone H3 is swapped.

4.   Are there any experiments using tissues obtained through Xenograft? To confirm that Compound C inhibits the proliferation effects should perform Ki67 staining on tumor tissue specimens from mice.

5.   In order to better understand the paper, it seems that a picture summarizing the effect of Compound C in Renca cells is needed. Please add a picture that summarizes the pathways mentioned in the paper.

Reviewer 2 Report

  The manuscript "Compound C inhibits Renca renal epithelial carcinoma growth in syngeneic mouse model by blocking cell cycle progression, adhesion and invasion" by Lee at al. aims to show the anticancer properties of CompC in RCC. In addition, authors have revealed the detailed molecular mechanism using various techniques. They have also deciphered the role of CompC in cell cycle arrest and important regulatory protein was investigated using western blot. Furthermore, authors have also shown how CompC concentration (, 2.5, 5, and 10 μM) affects the migration, migration and adhesion of Renca cells. In vivo tumor models have shown a really promising result when treated with CompC. This is a well designed study with proper control. The Introduction is nicely written and shows the aim of the study very clearly. The Methodology section is clearly defined and detailed in every aspect to reproduce the data. Results section has been described in detail and important outcomes have been highlighted. This study will definitely enhance the existing knowledge in the field diagnostic and treatment strategy for RCC. 

I do have one concern; as you have mentioned that Renca is procured from ATCC, as per ATCC cells should be cultured in RPMI-1640 then why did you use DMEM ? Changing media could adversely affect cell behavior, thus the whole study.  

Round 2

Reviewer 1 Report

revised well